# eWaterCycle: a hyper-resolution global hydrological model for river discharge forecasts made from open source pre-existing components

Rolf Hut<sup>1</sup>, Niels Drost<sup>2</sup>, Maarten van Meersbergen<sup>2</sup>, Edwin Sutanudjaja<sup>3</sup>, Marc Bierkens<sup>3,4</sup>, and Nick van de Giesen<sup>1</sup>

 <sup>1</sup>Chair of Water Resources Engineering, Faculty of Civil Engineering and Geosciences, Delft University of Technology, Delft, The Netherlands
 <sup>2</sup>Netherlands eScience Center, Amsterdam, The Netherlands
 <sup>3</sup>Department of Physical Geography, Utrecht University, Utrecht, The Netherlands

<sup>4</sup>Deltares, Utrecht, The Netherlands

Correspondence to: Rolf Hut (r.w.hut@tudelft.nl)

**Abstract.** eWaterCycle is an open source hyperresolution (10km x 10km) global hydrological forecasting framework that runs an ensemble of hydrological models. Forced with a weather forecast ensemble, it predicts river discharge and river discharge uncertainty nine days ahead. Daily satellite soil moisture observations are assimilated into the state of the model ensemble using an Ensemble Kalman Filter. We demonstrate that it is feasible to build such a system using pre-exisiting, open source,

- components that communicate through standard interfaces. The PCRGLOBWB2.0 (van Beek et al., 2011; Sutanudjaja et al., 2014) model is used to model hydrology globally, forced with GFS (Kanamitsu, 1989; Kanamitsu et al., 1991; Moorthi et al., 2001) weather forecast. The operational soil moisture product from the HSAF (Drusch et al., 2009; De Rosnay et al., 2011) service is assimilated into the model ensemble using OpenDA (Velzen et al., 2016), a data assimilation framework. Output of the model ensemble is presented in a Cesium (Analytical Graphics, 2011) based visualization. All communication be-
- tween framework components is through standard file types (NetCDF)(Rew and Davis, 1990) and services (Web Map Service) (de La Beaujardiere, 2006). Communication between model and data assimilation framework is through the Basic Model Interface (BMI)(Peckham et al., 2013). The forecasts is available at forecast.ewaterycle.org. By using standard open interfaces, the different components of the model can be replaced with relative ease, facilitating future model comparison studies without the need of extensive Computer Science support. This makes eWaterCycle, in addition to an operational forecasting model,
- a testbed environment where the impact of different model structures, input sources and / or data assimilation schemes can easily be studied. Setup instructions to run the eWaterCycle project on local hardware are provided, allowing the hydrological community to build on this open source framework.

## 1 Introduction

The need for a "hyper-resolution" global hydrological model has been expressed by Wood *et al* Wood et al. (2011) as a "grand
challenge" in hydrology. Global modeling of hydrology has improved in resolution to grids as small as 0.5° by 0.5° (Bierkens et al., 2015). The review by Bierkens (2015) of state of the art hydrological global models shows that global hydrological

models are used for climate sensitivity analysis (Nijssen et al., 2001), and for long term global change analysis (Prudhomme et al., 2014), but rarely for operational forecasting of river discharge, with Alfieri et al. (2013) a notable exemption.

On basin or continental scale, many hydrological ensemble prediction systems (HEPS) do exist (Thiemig et al., 2015; Kauffeldt et al., 2016; Emerton et al., 2016). Most HEPS take an ensemble of weather forecasts to force an ensemble of

<sup>5</sup> hydrological models. Data assimilation of observed states of the hydrological model is not a common feature of either global, or continental HEPS (Thiemig et al., 2015). For optimal estimation of hydrological variables, it is important to assimilate relevant observations (Lopez Lopez et al., 2015). Recent work by Kurtz et al. (2016) has shown that it is feasible to run very large scale, global HEPS on high performing computing infrastructure (super computers). This does, as Kurtz et al. (2016) states, require detailed knowledge on how to couple the data assimilation scheme to the model. Close collaboration between
10 hydrologists and computer scientists is, therefore, required.

The grand challenge as laid out by Wood et al. (2011) specifically argues that focus of the hydrological community should be on combining, or coupling, models from different subfields: surface and subsurface hydrology, land and atmosphere, water quantity and water quality. To achieve this, different models should be coupled using open interfaces, allowing researchers from different Earth science disciplines to work together and with computer scientists without having to know intimately each

- other's field. Different open interfaces to communicate between model(components) have recently been developed (Donchyts et al., 2010; Peckham et al., 2013). It should be possible that these models are written in different programming languages. For these coupled models to have societal value within the context of an HEPS, they should be run as an ensemble forecast, and use data assimilation. Data assimilation scheme and models should communicate through open interfaces, decoupling the need for computer scientists and geoscientists to have intimate knowledge of eachother's field, while still allowing geoscientists to
- reap the benefit of advances in the computer sciences.

In this paper we demonstrate that it is technically feasible to run a global hydrological ensemble forecasting system on high performance infrastructure without deep coupling. The resulting forecasts are available on forecast.ewaterycle.org. In our "eWaterCycle" system the hydrological model is coupled to the data assimilation scheme using an open interface. All interactions between the components: scheduler, preprocesser, data assimilation scheme, hydrological model and visualization are either through standard file formats or through open interfaces. The source code for all components of the eWaterCycle model is available in open source GitHub repositories, allowing anyone with access to the proper hardware to setup and run

their own global hydrological ensemble prediction system and, more importantly, to improve upon the system provided and make steps, as a community, to achieve the goals set out in the grand challenge.

#### 2 eWaterCycle architecture

- The eWaterCycle system consists of 5 components:
  - orchestrator: Cylc
  - pre- and post- processor: CDO

- data assimilation: OpenDA
- cross-language model coupling: Thrift
- model: PCR-GLOBWB 2.0
- visualization: Cesium and ncWMS
- 5 Furthermore, two input data sources are used: the GFS weather forecast and the EUMETSAT Soil moisture product. Figure 1 shows an overview of how the components and input sources are connected by the interfaces. All interfaces are either standard file formats (NetCDF and GRIB) or open interfaces (BMI and WMS). Tables 1 and 2 give an overview of all the components, input sources and interfaces, including where they can be found online. In the sections below every interface, input source and component will be described.

# 10 3 Input data sources

The eWaterCycle system has two main input sources that change daily: The GFS weather prediction and the EUMETSAT soil moisture product. Below we describe how these inputs are processed. The model does require more inputs, such as a Digital Elevation Model (DEM), but these do not change between model runs (see section 4.5 for details)

# 3.1 GFS

- The U.S. National Center for Environmental Prediction (NCEP) runs a deterministic weather prediction, the Global Forecast System (GFS)(Kanamitsu, 1989; Kanamitsu et al., 1991; Moorthi et al., 2001), as well as an ensemble of weather predictions, the Global Ensemble Forecast System (GEFS)(Toth et al., 2006; Zhu, 2005). The GFS has a higher spatial resolution (0.25° versus 1°) and is run for a longer forecasting time (16 days versus 8 days). The GEFS is run with 21 ensemble members. Other Numerical Weather Predictions are available, such as those provided by the European Centre for Medium-Range Weather
- Forecasts (ECMWF). eWatercycle uses the NCEP model because the data is available for anyone to download for free. A particular interesting feature of GFS and GEFS is the NOMAD online filter that can be used to extract and download only the variables of interest (Rutledge et al., 2006). For the PCR-GLOBWB 2.0 model these variables are precipitation and temperature. In eWaterCycle we create an input ensemble that is a hybrid of GFS and GEFS by superimposing the deviation of the GEFS ensemble mean on GFS. The script that downloads the data and produces the input ensemble for the PCRGlobWB 2.0 model
- is discussed in section 4.2.

# 3.2 ECWMF SM-DAS-2

The SM-DAS-2 product by ECWMF (Drusch et al., 2009; De Rosnay et al., 2011), available through the HSAF service, was chosen to be assimilated in eWaterCycle. In (Albergel et al., 2012) the data from SM-DAS-2 were compared to ground based

measurements of soil moisture. SM-DAS-2 is technically not 100% an observation, but rather the result of assimilating observations from the ASCAT satellite into the ECWMF land-surface model. From a computational point of view, SM-DAS-2 provides a sort of "worst case" scenario for eWaterCycle, as it is a large global high resolution observation dataset. By assimilating the SM-DAS-2 product into the PCRGlobWB 2.0 model it is shown that assimilating large datasets into an hydrological ensemble prediction system using open source components is feasible.

4 Components

The five components that make up the eWaterCycle system are described below. For each component the history and original purpose is briefly sketched followed by a description of the function the component has within the eWatercycle system.

# 4.1 Orchestration: Cylc

- Cylc is a workflow management system designed especially for running cyclic tasks such as our forecast, where each next forecast relies on results from the previous forecast (Oliver, 2008). Cylc tracks which tasks are able to run and automatically submits these to a cluster or supercomputer for processing. How then to perform a task is not up to Cylc. Instead user-provided commands or scripts are run. This makes it easy to integrate Cylc into an existing system, as most existing scripts can be used as is. Cylc was originally developed at the New Zealand National Institute of Water and Atmospheric Research (NIWA), and
- is now used at the UK Met Office and others.

In the eWaterCycle system, we use Cylc to automatically run our forecast, from downloading necessary weather data, to uploading our final result to the web application server. See Figure 3 for a graphical representation of our workflow by the Cylc monitoring application.

#### 4.2 Pre- and Post- processor: CDO

- The Climate Data Operators (CDO, 2015) are a set of command line operators that work on gridded data in NetCDF or GRIB files (see section 5.1 below). The CDO includes basic operations such as adding similar variables from different files with identical grids, but also more complex operators to interpolate gridded data to a new grid using different interpolation schemes (Kriging, nearest neighbor, bilinear, etc.). Multiple CDO operators can be linked into a single command, which will be executed efficiently in memory without saving the intermediate results. Multiple CDO commands can be combined together in bash-scripts, but integration in Python and other computing languages is also possible.

In eWaterCycle, CDO is used for pre-processing the weather forecast data into NetCDF files that the PCR-GLOBWB model (see section 4.5) uses as input. Furthermore, CDO is used to gather the outputs of the ensemble members, which each produce a single NetCDF file, into a single file that includes ensemble mean and standard deviation as input for the visualization (see section 4.6). The bash scripts used in eWaterCycle are available on GitHub, see table 2.

5

25

# 4.3 Data Assimilation: OpenDA

Data assimilation (DA), the science of incorporating state or output observations in continuously running forecasting models, is often built as an add-on to an existing model. However, implementing DA techniques both correctly and fast, in terms of computing time required, is no trivial matter (Kurtz et al., 2016). Also, research into improved DA schemes is hindered by this practice, as the improved scheme needs to be implemented for each model on which it needs to be tested. To solve this problem, OpenDA was developed (Velzen et al., 2016). OpenDA is a generic model independent data assimilation framework. Once an existing model has been coupled to OpenDA, it immediately has DA capabilities.

In the eWaterCycle system, we use OpenDA to assimilate soil moisture observations into our model using an Ensemble Kalman filter (EnKF) (Evensen, 2003). OpenDA creates 20 ensemble members of our model. The initial state of each member

- 10 (at T = -24 hours) is carried over from the previous forecast run. Each member is forced using input from each GEFS/GFS ensemble member as described in section 3.1. At T=0 the observations are assimilated into the model. As the amount of observations is large (about 50.000 pixels with observations), we use the sequential EnKF implementation as present in the OpenDA framework. This assimilates the observations one by one. This uses less memory, at the cost of an increased runtime and the additional assumption that all observations are independent. Having to use this workflow highlights that EnKF has
- scaling issues when large ensembles or large numbers of observations, such as satellite data, are assimilated. Although efficient memory variants of the EnKF exist (Hut et al., 2015), these have not yet been implemented in standard DA frameworks such as OpenDA. After the DA step, OpenDA runs our model for the remainder of the forcing to create a forecast. The state at T = 0after DA is also kept for the next forecast.

We have extended OpenDA with a model independent coupling based on Thrift and BMI. This *BMI Bridge* is capable of running any BMI compliant model, and can run multiple ensemble members in parallel on multiple machines. This version of

20 running any BMI compliant model, and can run multiple ensemble members in parallel on multiple machines. This version of OpenDA is currently available as part of the eWaterCycle repository on Github, see table 1. See sections 4.4 and 5.2 for more details.

Figure 2 gives an overview of the time dependencies in the eWaterCycle system. It shows how each day, observations are assimilated in the forecast for that day. The resulting assimilated state is the start (T = 0) for the next model run, resulting in a new forecast.

#### 4.4 Model Coupling: Thrift

Apache Thrift is a system for creating cross-language services (Slee et al., 2007). Thrift was originally developed by Facebook to couple services implemented using different programming languages, but is now hosted as an open source project by the Apache Foundation.

30 Thrift couples services by the use of a Remote Procedure Call (RPC) model. A programmer defines an interface, called service, in a language neutral fashion using a Interface Definition Language (IDL) file. From this file, Thrift then generates a client and server application for this service in a range of different programming languages. A service could, for instance, be implemented in Go, while a client is implemented in C++.

In the eWaterCycle system we use Thrift to couple our model (see Section 4.5) to OpenDA. Our model has been implemented in Python while OpenDA has been implemented in Java. Thrift allows us to bridge this language gap with very little effort. Furthermore, instead of implementing model specific functions, we use the BMI standard (see Section 5.2). This, in effect, creates a model independent bridge capable of coupling any model to OpenDA, written in any of the over 20 programming languages supported by Thrift.

# 4.5 Global Hydrological Model: PCR-GLOBWB 2.0

The global hydrological model PCR-GLOBWB (van Beek et al. (2011); Sutanudjaja et al. (2014), Sutanudjaja et al., in prep.) simulates continuous fields of fluxes and stocks in various water storage components, such as soil moisture, surface water, snow and groundwater on spatial and temporal scales, with spatial resolution of 5 arcminutes, or approximately 10 km at the equator.

PCR-GLOBWB computes, for each cell, the vertical water exchanges between soil layers and between top soil layer and atmosphere. Processes represented in the model are rainfall and snowmelt, percolation and capillary rise, as well as evaporation and transpiration fluxes. Under the soil layer is a groundwater store. Groundwater is fed by net groundwater recharge. The process of groundwater feding baseflow is modelled as a linear reservoir. This behaviour is extempt from direct influence

- of evaporation and transpiration fluxes. Capillary rise from the active groundwater store depends on the simulated groundwater storage, the soil moisture deficit and the unsaturated hydraulic conductivity. Fluxes are computed for various land cover types by including sub- grid variations. Differences in topography, vegetation phenology, and soil properties are included. PCR-GLOBWB includes a physically-based scheme for infiltration and runoff, resulting in direct runoff, interflow, as well as groundwater baseflow and recharge. River discharge is calculated by accumulating and routing all runoff components along the
- drainage network. For more details about the PCR-GLOBWB model, including its parameterization, we refer to its technical 20 reports and other relevant papers (Van Beek and Bierkens, 2009; Van Beek, 2008; Sutanudjaja et al., 2011, 2014). PCR-GLOBWB is used and described in other papers such as the recent Tangdamrongsub et al. (2016a, b).

Over time, several features of PCR-GLOBWB have been implemented since its first application (van Beek et al., 2011). One new feature concerns two-way interactions between water availability and abstraction, including simulating variations

- in irrigation and other water demands, such as livestock, industrial and domestic water demands, and allocation of available surface water and groundwater resources (see de Graaf et al. (2014); Wada and Bierkens (2014); Erkens and Sutanudjaja (2015)). Combined, these features determine how the demands are allocated to the available water resources and where the return flows of unconsumed water go. The new features lead to the simulation of essential feedbacks, particularly in irrigated areas where water demand is large. Together with many other new developments, these advanced and integrated water demand
- and allocation modules are being consolidated in the new version of PCR-GLOBWB 2.0. This new version fully integrates 30 global hydrology and water resource models and will be implemented at a spatial resolution of 5 arc minutes (Sutanudjaja et al., in prep.).

In this research, an alpha release of the 5 arc minutes version of PCR-GLOBWB 2.0 is used. In this alpha version all calculations are executed but irrigation and other human influences are situated within single pixels, as was the case in the

5

original one degree resolution model. Considering within pixel human influence is unrealistic at 5 arc minutes resolution and greatly reduced the accuracy of the simulated discharge. Furthermore, the irrigation routine requires historical data which is available when running PCRGLOBWB for historical analyses. We like to stress that the focus of this paper is not on the accuracy of the prediction but rather on the numerical and computational feasibility to create and run a hyper-resolution global hydrological model for river discharge forecasts made from pre-existing open source components.

Some other currently ongoing developments regarding PCR-GLOBWB include a coupling to lateral groundwater flow modelling for simulating spatio-temporal groundwater head dynamics ((Sutanudjaja et al., 2011, 2014; de Graaf et al., 2014; Sutanudjaja et al., 2014)). PCR-GLOBWB can be extended with an external surface water and flood routing scheme (Winsemius et al., 2013) and a physically-based freshwater surface temperature model (van Beek et al., 2012).

#### 10 4.6 Visualization: Cesium and ncWMS

ncWMS is a server side application capable of reading any NetCDF file written using the Climate and Forecasting (CF) conventions (Blower et al., 2013), and making the data available as a Web Map Service(WMS). ncWMS automatically determines available variables in a file, and creates maps colored according to map data and a user selected color scale. See Section 5.3 for more information on WMS.

Cesium is a Javascript 3D virtual Globe library (Analytical Graphics, 2011). It uses WebGL for rendering, which makes it very fast, and it is capable of displaying a wide variety of data types such as vectors, 3D models, and 2D maps.

To visualize the result of the forecast we have created Cesium-ncWMS, a web application based on ncWMS and Cesium. The forecast results are automatically uploaded to our web server running ncWMS. In turn, the web application can be used to change the settings for color maps and displayed data. The server uses the settings provided by the web application, together

with the data in NetCDF to provide WMS image tiles, time series data and legend graphics to the Cesium-NcWMS web application. The user can simultaneously zoom in to the very high resolution forecast results anywhere on the world, and get time series data for any point on the globe. See figure 4 for a screenshot.

#### **5** Interfaces

Interfaces are the glue between the components of the eWaterCycle system, which contains two data interfaces (BMI, WMS) and two data file formats or data abstractions (GRIB, NetCDF). Below the reasons for choosing the interfaces are explained.

#### 5.1 GRIB and NetCDF

Ideally different components should exchange data through interfaces if at all possible, reducing the need to write and read large files. However, many components still use files, both as inputs or outputs, as the main way of communicating with other components. In eWaterCycle two filetypes are used to share data between components: GRIB and NetCDF.

GRIB (GRIdded Binary or General Regularly-distributed Information in Binary form) is a data format used mostly to store weather data. The format was standardized by the World Meteorological Organization (WMO) (Dey and others, 2007). It has

been designed to store fields of gridded variables and has been the de-facto standard fileformat for numerical weather prediction data for decades.

The Network Common Data Form (NetCDF) is not so much a fileformat, as it is a data abstraction (Rew and Davis, 1990). NetCDF data can live on a server, which can be queried, or in a file, which can also be queried. In NetCDF, metadata are stored together with the data. The implementation of NetCDF makes it easy to query a NetCDF file and only receive the part of the file that is of interest. Variables, including their units, have to be precisely documented. Different fields of science have set standards on how to store the units relevant to their field. For geosciences, these are encoded in the Climate and Forecasting (CF) conventions (Eaton et al., 2003). The NetCDF software project is hosted by the University Corporation for Atmospheric Research (UCAR).

# 10 5.2 BMI

5

BMI (Basic Model Interface) is a standard for interfacing models in geosciences (Peckham et al., 2013). As the name implies, BMI asks very little of a model. BMI is explicitly designed to be easy to implement by modellers. BMI requires a model to implement a small number of functions, most usually already available in some form in a model. Examples include a function to initialize the model, get and set the state of variables, run the model one or more steps, and get some metadata on the model,

such as what variables are available, what units do these have, etc. Bindings for different languages, including Fortran, Python, Java, C, and C++, are available. All in all, implementing a BMI interface for a model should take no more than a day or so, far less than most other interfaces and frameworks.

In the eWaterCycle project, we use BMI as the interface between OpenDA and PCR-GlobWB. As these are written in different languages, we use Thrift to couple the two. The use of BMI also allows reuse of components in other projects. The OpenDA-BMI model bridge is already being re-used in the OpenStreams project (Weerts 2016, in preperation).

#### 5.3 WMS

Web Map Service (WMS) is a standard for retrieving map based image data from a remote service on the Internet (de La Beaujardiere, 2006). WMS allows a user to specify which part of the map is needed, which variables and styles need to be displayed, etc. WMS is implemented using a wide variety of backends, ranging from simple files to large database systems. WMS is widely supported in client applications, including both desktop applications, and 2D and 3D web applications.

In the eWaterCycle project we use WMS as the intermediary protocol between our visualization and the data server. As a proof of concept, we also created a second visualization making use of WMS. This was eWaterLeaf (Drost, 2016), a simple 2D map visualization showing the ensemble mean of the forecast of discharge based on Leaflet (Agafonkin, 2015). Both visualizations are re-usable in other projects. Our Cesium-ncWMS visualization makes use of some extensions to WMS present

in ncWMS, for instance to support a logarithmic value scale. To use Cesium-ncWMS with a different data server would therefor require some work.

10

#### 6 eWaterCycle hardware requirements

Although all software used and created in the eWaterCycle is available as open source, running the model also requires capable hardware. As the eWaterCycle forecast runs 20 ensemble members simultaneously, total storage, memory, and cpu requirements are dominated by these. In the setup used in the eWaterCycle system, our model produces around 5Gb of output, as well as 2Cb of state files needed for the next iteration of the forecast. For the antise members are dominated by the order of 150Cb storage is

5 as 2Gb of state files needed for the next iteration of the forecast. For the entire ensemble, on the order of 150Gb storage is required per forecast. This can be reduced by producing less variables in the output, or increased by keeping more intermediate data for verifying the correctness of the model and data assimilation.

As for memory and cpu requirements, each model instance requires 20Gb of memory, with short peaks of about 40 Gb. We have implemented the capability to spread the ensemble members over multiple machines. For computing the live forecast we use 3 machines (nodes) of the Dutch Cartesius supercomputer with 256Gb memory each.

The website platform we use has more modest requirements. We use a t2.medium instance in the Amazon EC2 Cloud to host the website and data server. This has 2 (virtual) cores and 4Gb of memory. When under heavy load, EC2 will automatically start another instance of the service, and load balance requests between the two. The website can be visited at forecast.ewaterycle.org (van Meersbergen et al., 2015).

## 15 7 Preliminary validation and opportunities for further development

The focus of this paper is on the development and technology of the eWaterCycle system and not on its accuracy. A thorough statistical evaluation and error attribution of the output of eWaterCycle will be the focus of a different paper. A brief preliminary validation is provided here to give the readership a first glance at the output of eWaterCycle and its general validity.

Four discharge stations from the Global Runoff Database (GRDB) as collected by the Global Runoff Data Center (GRDC
in the Bundesanstalt für Gewässerkunde, 2015) were selected. The selection was based on the criteria data availability and global coverage. Only stations for which daily discharge measurements were available for a significant part of 2015 were selected. Geographical coverage was optimized by choosing rivers on four continents, Europe, North America, South America and Africa. No Asian rivers met our criteria. See table 3 for meta-data on the selected stations.

- For the locations of the four stations, the discharge as predicted by eWatercycle has been received from the eWaterCycle archive for every forecast made in the period 2015-04-01, when we first started eWaterCycle, until 2015-12-31, the last day for which discharge is available in the GRDC database at the time of writing. From each of these forecasts, the first and the last forecast day is selected. The four plots in figure 5 show the measured discharge at the stations, as well as the predicted ensemble for the first day and the ninth of the forecast of eWaterCycle. The Nash-Sutcliffe coefficient between the ensemble mean and the observed discharges for both the first day forecast as well as the ninth day forecast are provided in table 3.
- 30 Figure 5 and table 3 clearly show that the current alpha version of the model does not forecast river discharge too well, as was to be expected from the description of the model in section 4.5. Especially rivers with large human influence, such as the Colorado, perform rather poorly. The forecasts for the larger rivers (Zambezi and Colorado) slowly converge towards realistic discharge values, indicating that a longer spin-up is required for PCR-GLOBWB 2.0. Figure 5 and table 3 clearly show that for