# Peer review of "eWaterCycle: a hyper-resolution global hydrological model for river discharge forecasts made from open source pre-existing components"

_Geoscientific Model Development, 2016_

## Short Comment (SC1) · 24 Oct 2016

Dear authors,

In my role as Executive editor of GMD, I would like to bring to your attention our Editorial version 1.1:

http://www.geosci-model-dev.net/8/3487/2015/gmd-8-3487-2015.html

This highlights some requirements of papers published in GMD, which is also available on the GMD website in the 'Manuscript Types' section:

http://www.geoscientific-model-development.net/submission/manuscript_types.html

In particular, please note that for your paper, the following requirement has not been met in the Discussions paper:

- "The main paper must give the model name and version number (or other unique identifier) in the title."

Please add a version number or other unique identifier for eWaterCycle in the title upon your revised submission to GMD.

Yours,

Astrid Kerkweg

---

## Author Comment (AC1) · 21 Nov 2016

dear executive editor,

thank you for pointing out our error in not including the version number in the title. In the final revised document, upon publishing in GMD, we will add the version number of the GitHub repo that holds all our code.

thanks for your comment,

dr. ir. Rolf Hut

---

## Referee Comment (RC1) · Anonymous Referee #1 · 7 Dec 2016

General comment

The manuscript describes a new computationally advanced global hydrological forecasting system that combines a number of pre-existing open-source components. It addresses relevant scientific modelling questions within the scope of GMD and EGU, and provides a practical method (and freely available tools) how freely available ensemble weather forecasts can be used as input to a global hydrological model, how to include data assimilation and how to automatically visualize the modeling results on a webserver. Therefore, I think the manuscript is certainly very interesting to hydrologists who lack knowledge on available computer science methods and tools. However, as also stated by the authors, the system is currently not able to produce reliable 9

day forecasts of river discharge or even estimate river discharge at the current day (as compared to observations). Unfortunately, the presentation of the work is very "sloppy" and requires an intensive revision. On the one hand, typos and other formal/language errors abound and requires careful language editing. More importantly is the lack of thoroughness and clarity in the description of the work. This strongly decreases the knowledge gain of the reader.

Specific comments

1) The authors call their product "eWaterCycle" alternatively a "global hydrological model for river discharge forecasts" (in the title) or a "global hydrological forecasting framework" (in the first line of the abstract). I may suggest calling it a "global hydrological forecasting system".

2) In my opinion, it is not appropriate to use the term "hyper-resolution" as in Wood et al. 2011, hyper-resolution refers to a resolution of 1 km x 1 km, but not, as here, 10 km x 10 km. In addition, the climatic drivers are only available at a resolution of approx. 25 km x 25 km. Due to 1) and 2), I think that the title is not adequate.

3) In the first line of the abstract, it is stated that eWaterCycle runs an ensemble of hydrological models. This seems to be incorrect, as just global hydrological model is included according to the manuscript for providing the forecasts available at forecast.ewatercycle.org.

4) P2L16-18: Why is data assimilation necessary for having societal value? And: it is tautological that "in the context of an HEPS, they should be run as ensemble forecasts".

5) Section 3.1., in particular P2L23: explain better how the input ensemble was generated, it is not clear what "superimposing the deviation of the GEFS ensemble mean on GFS" might mean. How many ensemble members are used as input to the 20 realizations of PCR-GLOBWB? (in Section 4.2, there is a reference to 3.1. but I cannot find the information in 3.1) What is the temporal resolution of the climate input?

6) Section 3.2.: The "observational" product that is assimilated needs to be described in more detail, including the spatial and temporal resolution, and that it represents soil moisture down to which depth beneath the land surface.

7) Section 4.2: The data assimilation procedure has to be explained better relating it to both the hydrological model (soil moisture of which layer) and the "observational" data.

8) Section 4.5: Here, a focus should be on modeling of the soil water balance as soil moisture is assimilated. Also, information on the assumed soil moisture modelling errors required for data assimilation needs to be provided. The last paragraph of the section should be deleted as it is not pertinent to the manuscript topic.

9) Figure 2 is hard to read as the sizes for the letters vary too much. Use larger size for letters in Fig. 3, too.

10) Figure 4: The plume is shown with two colours. Please explain their meaning not only in the caption but also on the website. Besides, in case of smaller river discharge, the plume can show negative discharge values. You need to point out somewhere in the paper that the Cesium-ncWMS visualization currently leads to this wrong visualization.

11) What is missing is a discussion of the results in the light of other hydrological forecasting systems both at the global scale (e.g. GloFAS) and a the basin scale. For a scientific paper on a new forecast system, it is paramount to relate to previous experiences with such systems, e.g. comparing computer methods used within the systems but also discussing the accuracy of forecasts.
* * *

---

## Referee Comment (RC2) · Anonymous Referee #2 · 5 Jan 2017

Within this manuscript the authors describe the development of a global hydrologic forecasting framework (eWaterCycle). This model is designed around several open source components that are used to predict global river discharge 9 days ahead at a 10km x 10km resolution. The development of such a system, which is composed entirely of open source components and freely available data does present a beneficial contribution to GMD and the scientific community, as it will allow hydrologists and modelers from a wide variety of specific research areas to focus on implementing different model components of relevance to them. As described in this paper, the model represents a useful framework for creating these coupled systems. It does not, as the authors acknowledge, represent a good forecasting system for global discharge (negative NSE for all nowcasts, as described in Section 7) and further work is clearly needed to improve the model performance. With regards to the manuscript itself, the organization and level of detail in describing model components is lacking and as such, it is hard for the reader to fully grasp the specifics on how this model has been constructed (which is seemingly the most important component of this work). In addition to this, the manuscript would benefit from significant editing to correct spelling and grammatical errors throughout.

Specific Remarks:

1) As described, this model should not be classified as hyper-resolution as presented by Wood et al. (2011). At a resolution of 10km, this would fall into the category of the coarse resolution models described in that paper (with 1km x 1km being hyper-resolution). In addition to this, the model forcings used are coarser than the model resolution (~25km x 25km). I think that this term should be removed from the title and the text.

2) What are spatial and temporal resolutions of the forcings used (Section 3)? What methods are used to handle the differences between products? (Presumably this should be covered by the CDO section (4.2))

3) More details are needed about the ensemble forcings used (Section 3.1). Line 23, Page 3 mentions "superimposing the deviation of the GEFS ensemble mean on GFS". It is not clear what this means and the reference to this in Section 4.3 is circular, offering no additional information. For a model with a large focus on using an EnKF, it is important that the reader understands how these ensemble members are being generated.

4) In Section 3.2, more details are needed for the assimilated data product (spatial and temporal resolution as above). In Line 3, Page 4 it should be clarified that this data product represents the "worst case" from the perspective of computational intensity, as this is a "best case" scenario from the perspective of global earth observations.

5) In Section 4.3, a description of the data assimilation system is presented but it is not clear how this system interacts with the hydrologic model. In this instance, soil moisture is being assimilated but it is unclear whether other variables could be assimilated in the same system. If so, what model parameters/states are being adjusted by the assimilation scheme? Given the claim that this model allows for the substitution of hydrologic models and assimilations schemes (Page 1 Line 15), this is an important aspect to clarify and explain further. It may also be beneficial to describe the hydrologic model before this section to give some context for the assimilation system.

6) The last paragraph of Section 4.5 does not seem relevant and I believe it should be removed. In addition, Section 4.5 would benefit from further explanation of the specific model processes of relevance to the problem rather than the history of the model (i.e. which specific model processes and innovations are used and why was the model chosen).

7) How does this modeling framework compare to other similar systems such as the Global Flood Awareness System (GloFAS) from ECMWF, the National Water Model (NWM) from NOAA, or the Global Flood Monitoring System (GFMS) from University of Maryland (or others)? I think it is very important to include a discussion on this to provide some context for the work and illustrate the importance of open source model components, as used here. Without this, it is hard to say that this manuscript meets the GMD criteria of being a "substantial advance in modelling science".

8) In Section 7 the authors present the results of limited validation work (acknowledging that accuracy is not the goal of this paper). Despite this, I believe that the authors should address the very poor performance further (almost all NSEs are negative). Are these errors entirely attributed to the individual components of this system (model parameters, forcings etc.) or is there an issue with the coupling between each of these components (the main goal of this work)? When all the validation results are this poor, it is hard to believe that the model functions correctly and passes the test of credibility (whether that is true or not). Further validation or analysis is needed to illustrate that

these are issues with the model components and not the coupling framework. Other metrics for validation such as the correlation or bias might also be useful to readers.

9) Figure 3 is hard to read with small fonts and different colored backgrounds. Consider increasing the size of the text and the shapes. A key for the colors might also be useful.